SciPost Physics Proceedings

Submission

# The strong coupling from $\;e^+e^- \to$ hadrons

D. Boito[1], M. Golterman[2,3]*, A. Keshavarzi[4†], K. Maltman[5,6],
D. Nomura[7], S. Peris[3], T. Teubner[4]

**1** Instituto de Física de São Carlos, Universidade de São Paulo,
CP 369, 13570-970, São Carlos, SP, Brazil
**2** Department of Physics and Astronomy, San Francisco State University
San Francisco, CA 94132, USA
**3** Department of Physics and IFAE-BIST, Universitat Autònoma de Barcelona
E-08193 Bellaterra, Barcelona, Spain
**4** Department of Mathematical Sciences, University of Liverpool,
Liverpool L69 3BX, U.K.
**5** Department of Mathematics and Statistics, York University
Toronto, ON Canada M3J 1P3
**6** CSSM, University of Adelaide, Adelaide, SA 5005 Australia
**7** KEK Theory Center, Tsukuba, Ibaraki 305-0801, Japan
*maarten@sfsu.edu
†Currently at the University of Mississippi (email: aikeshav@olemiss.edu)

November 6, 2018

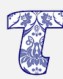 *Proceedings for the 15th International Workshop on Tau Lepton Physics,
Amsterdam, The Netherlands, 24-28 September 2018*
scipost.org/SciPostPhysProc.Tau2018

## Abstract

We use a new compilation of the hadronic $R$-ratio from available data for the process $e^+e^- \to$ hadrons below the charm mass to determine the strong coupling $\alpha_s$, using finite-energy sum rules. Quoting our results at the $\tau$ mass to facilitate comparison to the results obtained from similar analyses of hadronic $\tau$-decay data, we find $\alpha_s(m_\tau^2) = 0.298 \pm 0.016 \pm 0.006$ in fixed-order perturbation theory, and $\alpha_s(m_\tau^2) = 0.304 \pm 0.018 \pm 0.006$ in contour-improved perturbation theory, where the first error is statistical, and the second error combines various systematic effects. These values are in good agreement with a recent determination from the OPAL and ALEPH data for hadronic $\tau$ decays. We briefly compare the $R(s)$-based analysis with the $\tau$-based analysis.

## 1  Introduction

Recently, new compilations of data for the $R$-ratio $R(s)$, measured in the process $e^+e^- \to$ hadrons$(\gamma)$, have appeared, mostly motivated by the aim to improve the dispersive prediction for the hadronic vacuum polarization part of the muon anomalous magnetic moment [1–3]. As $R(s)$ is directly proportional to the electromagnetic (EM) QCD vector spectral function, it also gives access to other QCD quantities of interest. One of those is the strong coupling $\alpha_s$, which can be extracted from $R(s)$ using finite-energy sum rules (FESRs) similarly to the extraction of $\alpha_s$ from the QCD spectral functions measured in hadronic $\tau$ decays.

The extraction of $\alpha_s$ from $R(s)$ is interesting because it provides us with an alternative determination of the strong coupling from data at relatively low energies, thus providing another direct test of the running of the strong coupling as predicted by perturbation theory. It can also directly be compared with the determination from hadronic $\tau$ decays. In this talk, we give a brief overview of the determination of $\alpha_s$ from $R(s)$, summarizing Ref. [4], to which we refer for details.

## 2  A new compilation of $R$-ratio data

The data set we employed for our work is that of Ref. [2], and it is shown in the left panel of Fig. 1. This plot shows $R(s)$ as a function of the square of the center-of-mass energy $s$, in GeV$^2$, below the threshold for charm production. At large $s$, $R(s)$ is expected to approach the parton-model value $R = 2$, plus small perturbative corrections.

In the right panel of Fig. 1 we show a blow-up of these same data, for 2 GeV$^2 \leq s \leq 6$ GeV$^2$. This plot shows more clearly that there are a lot more data in the region $s \leq 4$ GeV$^2$, where $R(s)$ was compiled from summing exclusive-channel experiments, than in the region $s \geq 4$ GeV$^2$, where $R(s)$ was compiled from inclusive experiments.[1] This implies that an extraction of $\alpha_s$ using all data below 4 GeV$^2$ will yield a value with a much smaller error than an extraction of $\alpha_s$ from $R(s)$ for a value of $s$ where QCD perturbation theory directly applies. As will be explained in the next section, FESRs provide us with the tool to use all data above threshold $(s = m_\pi^2)$.

## 3  Finite energy sum rules

We consider the EM vacuum polarization $\Pi(z = q^2)$, and integrate its product with a polynomial weight function $w(z/s_0)$ along the contour shown in Fig. 2, where the circle has radius $s_0$. $\Pi(q^2)$ is analytic everywhere in the complex $q^2$ plane except along the

---

[1]For a much more detailed description and discussion of the compilation we refer to Refs. [2,4].

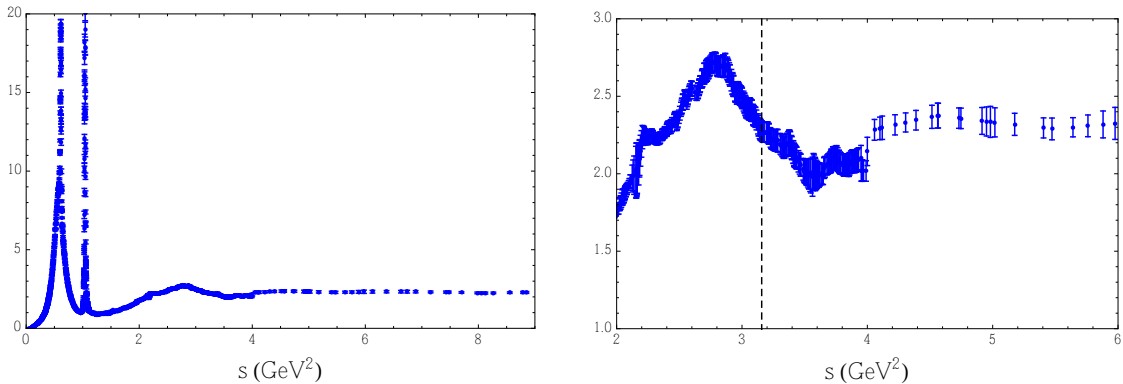

Figure 1: *Left panel: R-ratio data from Ref. [2], as a function of $s$, the hadronic invariant squared mass. The three-flavor, massless parton-model value is 2. Right panel: A blow-up of the region $2 \leq s \leq 6$ GeV$^2$.*

positive $q^2$ axis, and Cauchy's theorem thus implies that the integral around this contour vanishes. Splitting the integral into one part along the circle, and one part going back below and forth above the positive axis, and using that

$$\rho(s) = \frac{1}{\pi} \text{Im}\, \Pi(s) = \frac{1}{2\pi i} \left( \Pi(s + i\epsilon) - \Pi(s - i\epsilon) \right) , \tag{1}$$

we find, using also $\rho(s) = \frac{1}{12\pi^2} R(s)$, the FESR with weight $w$

$$I^{(w)}(s_0) \equiv \frac{1}{s_0} \int_{m_\pi^2}^{s_0} ds\, w(s/s_0)\, \frac{1}{12\pi^2}\, R(s) = -\frac{1}{2\pi i s_0} \oint_{z=|s_0|} dz\, w(z/s_0)\, \Pi(z) . \tag{2}$$

In this equation, the left-hand side represents the "data" side, and it incorporates all data between threshold and $s = s_0$. The right-hand side represents the "theory" side, and, if $s_0$ is large enough, perturbation theory should provide a good representation of the theory.

In more detail, if $s_0$ is large enough, we can use the theory representation

$$\Pi(z) = \Pi_{\text{pert}}(z) + \Pi_{\text{OPE}}(z) + \Pi_{\text{DV}}(z) . \tag{3}$$

The first term, $\Pi_{\text{pert}}(z)$, is obtained from massless perturbation theory, and is known to order $\alpha_s^4$ [6]. The OPE (operator product expansion) part can be parametrized in terms of the "condensates" $C_{2k}$ as

$$\Pi_{\text{OPE}}(z = q^2) = \sum_{k=1}^{\infty} \frac{C_{2k}}{(-q^2)^k} , \tag{4}$$

while the "duality-violation" part $\Pi_{\text{DV}}(z)$ represents contributions to $\Pi(z)$ manifested by the presence of resonance peaks, which are not captured by perturbation theory or the OPE. The $D = 2k = 2$ term in the OPE corresponds to the mass corrections that can be calculated in perturbation theory, and is thus known. Condensates with $D = 2k > 2$ are not known, and will be treated as free parameters in our fits.[2]

---

[2] In reality, the coefficients $C_{2k}$ are logarithmically dependent on $q^2$. However, this dependence can be safely neglected in the application of FESRs to the $R(s)$ data, given their precision, at least for $k > 1$ (we do take this $q^2$ dependence into account for $k = 1$). Likewise, the up and down quark masses can be safely set equal to zero, and $C_2$ can thus be expressed in terms of the strange quark mass $m_s(q^2)$ and $\alpha_s(q^2)$.

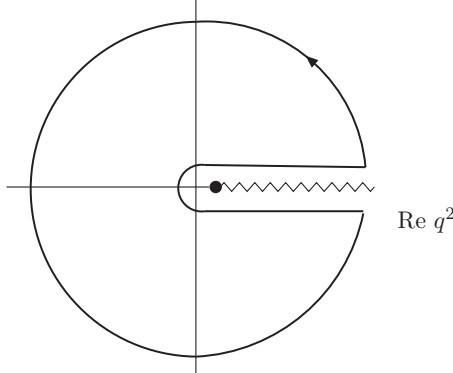

Figure 2: *Analytic structure of $\Pi(q^2)$ in the complex $z = q^2$ plane. There is a cut on the positive real axis starting at $s = q^2 = m_\pi^2$ (see text). The solid curve shows the contour used in Eq. (2).*

In our analysis, we will employ the weight functions

$$
\begin{aligned}
w_0(y) &= 1 \,, \\
w_2(y) &= 1 - y^2 \,, \\
w_3(y) &= (1-y)^2(1+2y) \,, \\
w_4(y) &= (1-y^2)^2 \,.
\end{aligned}
\tag{5}
$$

From Eq. (2), one sees that, apart from $C_2$, $C_6$ contributes to $I^{(w_2)}$, $C_6$ and $C_8$ contribute to $I^{(w_3)}$, and $C_6$ and $C_{10}$ contribute to $I^{(w_4)}$. We avoid weights with a term linear in $y$, as it was argued that perturbation theory for such weights should be expected to have poor convergence properties [5]. In our analysis, we also included EM corrections to perturbation theory.

Duality violations, represented by the term $\Pi_{\rm DV}(z)$, are expected to give a contribution which decreases with increasing $s_0$. In addition, their largest contribution to the integral on the right-hand side of Eq. (2) is expected to come from the part of the circle closest to the real axis, *i.e.*, $z \approx s_0$. Their contribution is thus suppressed for $w = w_2$, which has a single zero at $z = s_0$ ($w_2$ is "singly pinched"), and more suppressed for $w = w_{3,4}$, which both have a double zero at $z = s_0$ ($w_{3,4}$ are "doubly pinched").

Let us compare the experimental values of $I^{(w)}(s_0)$ for the four weights (5) with the value of $R(s_0)$ itself, for example at $s_0 = 4$ GeV$^2$, evaluated on the data of Ref. [2]. We show these values in the following table:

| quantity | OPE coefficients: $D = 2k$ | error at $s_0 = 4$ GeV$^2$ |
|:---:|:---:|:---:|
| $R(s_0)$ | $-$ | 4.3% |
| $I^{(w_0)}(s_0)$ | $D = 2$ | 1.04% |
| $I^{(w_2)}(s_0)$ | $D = 2,\ 6$ | 0.73% |
| $I^{(w_3)}(s_0)$ | $D = 2,\ 6,\ 8$ | 0.56% |
| $I^{(w_4)}(s_0)$ | $D = 2,\ 6,\ 10$ | 0.59% |

We see immediately that the spectral moments $I^{(w_{0,2,3,4})}(s_0)$ are known to a much higher precision than $R(s_0)$ from the same data. The reason is of course that the spectral

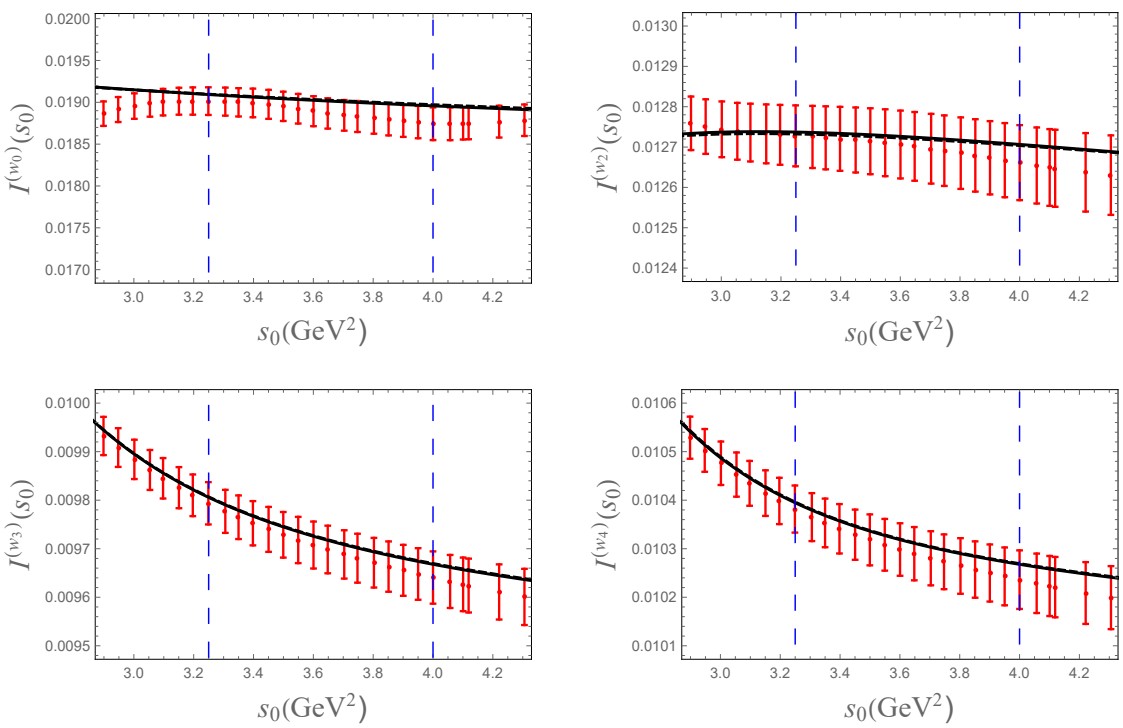

Figure 3: *Comparison of the data for $I^{(w)}(s_0)$ with the fits on the interval $s_0^{\min} = 3.25$ to 4 GeV$^2$, for $w = w_0$ (upper left panel), $w = w_2$ (upper right panel), $w = w_3$ (lower left panel), and $w = w_4$ (lower right panel). Solid black curves indicate FOPT fits, dashed curves CIPT. The fit window is indicated by the dashed vertical lines.*

moments include all data for $R(s)$ from threshold to $s = s_0$. Similar observations apply to values of $s_0$ other than 4 GeV$^2$. This explains why the application of FESRs to the $R$-ratio data leads to a much more precise determination of $\alpha_s$ from these data than a direct determination from $R(s)$ at a fixed value of $s$.

## 4    Results

We now summarize the results of our fits of the FESRs (2) to the data. Our fits were carried out on a window $s_0 \in [s_0^{\min}, s_0^{\max}]$, with 3.25 GeV$^2 \leq s_0^{\min} \leq 3.80$ GeV$^2$ and $s_0^{\max} = 4$ GeV$^2$, finding good stablity for these values of $s_0^{\min}$. In Fig. 3 we show typical fits for all four weights (5), with $s_0^{\min} = 3.25$ GeV$^2$. Fits were carried out neglecting the duality-violating term $\Pi_{\text{DV}}$ in Eq. (3). All fits are correlated, and have $p$-values varying from 0.09 to 0.42.

We note that the values of $s_0$ used in our fits are all larger than the square of the $\tau$ mass $m_\tau^2$, the kinematic end point for a similar analysis of spectral functions measured in hadronic $\tau$ decays. In particular, we notice that in the $e^+e^-$ case good fits are obtained neglecting duality violations, in contrast to the $\tau$-decay case (see Sec. 5 below). For $w = w_0$, a remnant of integrated duality violations (the small oscillation) is visible, but the fit is consistent with the data, visually, and as confirmed by the quality of the fit. For

higher weights, all of which involve pinching, no effect from integrated duality violations is visible at all.

We used two different resummations of the perturbative series commonly employed in such sum-rules analyses, FOPT (fixed-order perturbation theory) and CIPT (contour-improved perturbation theory [7]). For a more detailed discussion, we refer to Refs. [4,5,8] and references therein, as well as Refs. [9,10].

In the table below, we show our results for the values of $\alpha_s(m_\tau^2)$ obtained from these fits, where we quote $\alpha_s$ at the $\tau$ mass in order to facilitate comparison with values obtained from hadronic $\tau$ decays:

| weight | $\alpha_s(m_\tau^2)$ (FOPT) | $\alpha_s(m_\tau^2)$ (CIPT) |
|--------|------------------------------|------------------------------|
| $w_0$  | 0.299(16)                    | 0.308(19)                    |
| $w_2$  | 0.298(17)                    | 0.305(19)                    |
| $w_3$  | 0.298(18)                    | 0.303(20)                    |
| $w_4$  | 0.297(18)                    | 0.303(20)                    |

Clearly, there is excellent agreement between the values obtained from different weights. This agreement is also found for the fit values for $C_6$, between the weights $w_2$, $w_3$ and $w_4$ [4]. The errors shown are a combination of the fit error and the error due to the variation of $s_0^{\min}$, where the first error dominates the total error.[3]

In Ref. [4] we carried out a number of additional tests. First, we did a number of fits with $s_0^{\max}$ or both $s_0^{\min}$ and $s_0^{\max}$ in the inclusive region $s > 4$ GeV$^2$. We found results consistent with those reported in the table above but including data in the inclusive region does not lead to a reduction of the errors shown in the table.

Second, while fits without duality violations lead to good $p$-values, we tested the stability of the fits with weight $w_0$ against the inclusion of a model for duality violations. We chose the weight $w_0$ as it is the weight which is most sensitive to duality violations, with no pinching at $z = s_0$. The model we used is described in detail in Ref. [4], uses input for the $I = 1$ channel from $\tau$ decays [4,11], and is based on theoretical insights about duality violations developed in Ref. [12] (and references therein).

Figure 4 shows a summary of this analysis for the FOPT case.[4] Colored data points (diamonds and squares) show fit values of $\alpha_s(m_\tau^2)$ as a function of $s_0^{\min}$ (with $s_0^{\max} = 4$ GeV$^2$; see figure caption for details). These are fits which do not include duality violations, $i.e.$, $\Pi_{\mathrm{DV}}(z)$ in Eq. (3) is omitted in these fits. The purple horizontal line shows the average value $\alpha_s(m_\tau^2) = 0.298$, and the dashed horizontal lines show the values $0.298 \pm 0.005$ with the 0.005 representing the fluctuation in $\alpha_s(m_\tau^2)$ values we find when we let $s_0^{\min}$ vary between 3.25 and 3.80 GeV$^2$.[5]

The black filled circles represent fit values for $\alpha_s(m_\tau^2)$ from fits with weight $w_0$ which do include duality violations. Two observations can be made: First, fits for smaller values of $s_0^{\min}$ than 3.25 GeV$^2$ become more stable; $p$-values for fits with $s_0^{\min} < 3.25$ GeV$^2$ become much larger and acceptable, while $p$-values for $s_0^{\min}$ values between 3.25 and 3.80 GeV$^2$ range between 0.20 and 0.50. Second, the central values for $\alpha_s(m_\tau^2)$ with $s_0^{\min}$ between 3.25 and 3.80 GeV$^2$ are completely consistent with the estimate $\alpha_s(m_\tau^2) = 0.298 \pm 0.005$. We conclude that indeed our fits are stable with respect to the inclusion of duality violations, and thus that they can be ignored within current errors in the analysis based on the

---

[3]Another error, due to the fact that the $O(\alpha_s^5)$ term in perturbation theory is unknown, is negligibly small.

[4]The CIPT case is very similar.

[5]Note that this is not the full error bar on $\alpha_s(m_\tau^2)$, because it does not include the fit error. The total error on fit values for $\alpha_s(m_\tau^2)$ is much larger.

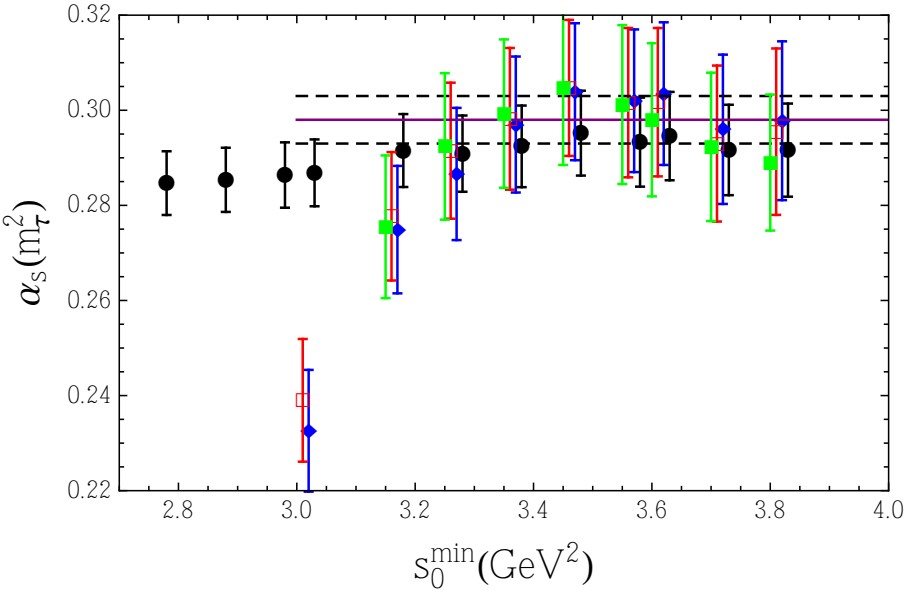

Figure 4: *The FOPT strong coupling $\alpha_s(m_\tau^2)$ as a function of $s_0^{\min}$. Blue data points (diamonds) represent values of $\alpha_s(m_\tau^2)$ from fits with weight $w_0$, red (open squares) those from fits with weight $w_2$, green (filled squares) those from fits with weight $w_3$, all without inclusion of duality violations. Black data points (filled circles) correspond to the values from fits with weight $w_0$, with the inclusion of duality violations. The solid, purple horizontal line shows the value 0.298, with the dashed horizontal lines showing the values $0.298 \pm 0.005$. The red, blue and black data points have been slightly offset horizontally for visibility.*

$R$-ratio, which allows us to probe values of $s_0$ significantly larger than $m_\tau^2$. We refer to Refs. [4, 12, 13] for more detailed recent information on the theory and role of duality violations.

## 5  Difference with determination from hadronic $\tau$ decays

In this section, we present a brief comparison between FESR fits of moments of the non-strange $I = 1$ vector spectral function obtained from hadronic $\tau$ decays [14], and FESR fits of the EM spectral function proportional to $R(s)$. As we will see in more detail in the next section, values for $\alpha_s(m_\tau^2)$ are consistent between the two cases. Here, instead, we compare the fits themselves.

Figure 5 shows fits of the moments $I^{(w_0)}(s_0)$ (upper panels) and $I^{(w_2)}(s_0)$ (lower panels), comparing these fits between fits based on the $\tau$ data (left panels) and fits based on the $e^+e^-$ data (right panels). The $\tau$-based fits have $s_0^{\max} = m_\tau^2$ and $s_0^{\min} = 1.55$ GeV$^2$; the $e^+e^-$-based fits have $s_0^{\max} = 4$ GeV$^2$ and $s_0^{\min} = 3.25$ GeV$^2$. In the $\tau$ panels, the blue curve represents FOPT fits with duality violations and the red dashed curve CIPT fits with duality violations. The black curves represent the perturbation theory plus OPE parts of these fits, omitting the duality-violating part. In the $e^+e^-$ panels, which just reproduce the top panels already shown in Fig. 3, the black curves represent FOPT (solid)

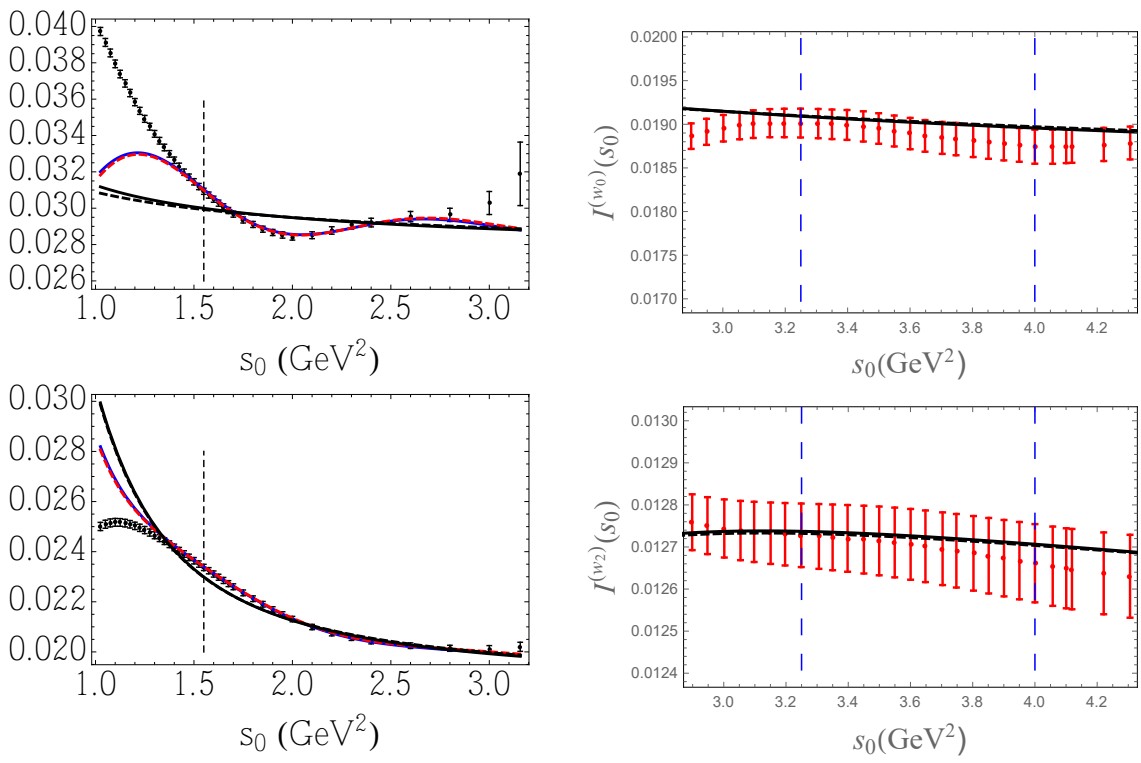

Figure 5: *Comparison of FESR fits extracting $\alpha_s$ from hadronic $\tau$ data (left panels) vs. $e^+e^- \to$ hadrons($\gamma$) (right panels). Top panels show fits with weight $w_0$, bottom panels show fits with weight $w_2$. For more detail, see main text.*

and CIPT (dashed) fits, with no duality violations.

Duality violations show up in the data points as oscillations around the perturbation theory plus OPE curves (black solid and dashed curves in all panels). Clearly, duality violations are very visible in the left panels. In contrast, they are barely visible in the upper right panel, and not visible in the lower right panel. These comparisons of theory with data show that duality violations cannot be ignored in the $\tau$-based results, while fits of moments of $R(s)$ at sufficiently higher $s_0$ are consistent with integrated duality violations being small enough at these higher values to be neglected, within current errors. This is consistent with the expected exponential decay of the duality-violating part of the spectral function with increasing $s$, as discussed in more detail in Ref. [13].

# 6    Final results and conclusions

Our final results for $\alpha_s(m_\tau^2)$ from the FESR-analysis of $R(s)$ is

$$
\begin{aligned}
\alpha_s(m_\tau^2) &= 0.298(17) \quad \text{(FOPT)} , \\
&= 0.304(19) \quad \text{(CIPT)} .
\end{aligned}
\tag{6}
$$

We note that the error is dominated by the fit errors, obtained by propagating the errors on the data compilation of Ref. [2]. This can be directly compared with the values obtained from the $\tau$-based analysis [11]:

$$\begin{aligned}
\alpha_s(m_\tau^2) &= 0.303(9) \qquad \text{(FOPT)} , \\
&= 0.319(12) \qquad \text{(CIPT)} .
\end{aligned} \tag{7}$$

There is excellent agreement between the results obtained from $e^+e^-$, and those obtained from $\tau$ decays. We note the much reduced difference between the FOPT and CIPT values in the $e^+e^-$ analysis, which we believe can be partially ascribed to the fact that these values are extracted from spectral-weight moments at larger $s_0$, where the convergence properties of perturbation theory are expected to be better.

We also quote the $e^+e^-$-based values after running the values of Eq. (6) to the $Z$-mass, converting from three to five flavors:

$$\begin{aligned}
\alpha_s(m_Z^2) &= 0.1158(22) \qquad \text{(FOPT)} , \\
&= 0.1166(25) \qquad \text{(CIPT)} .
\end{aligned} \tag{8}$$

These values are both consistent, within errors, with the world average as reported in Ref. [15], confirming the running predicted by QCD between the scale of the $e^+e^-$ analysis and $M_Z$ [16].

Finally, we point out that the $R$-ratio data can be used to test results obtained in the $\tau$-based approach. Any strategy employed in the application of FESR-based fits to the spectral moments obtained from hadronic $\tau$ decays can be applied to similar spectral moments obtained from $R(s)$, limiting oneself to the kinematic regime allowed by the $\tau$ data, $i.e.$, $s_0 \leq m_\tau^2$. Clearly, the match between theory and data should then also work above the $\tau$ mass; in fact, if anything, it should be better. We have applied this test to the "truncated OPE" strategy employed by Refs. [14, 17], finding that there are very serious, systematic problems with that approach. This confirms the conclusions of Ref. [18]. For a preliminary overview of this analysis, we refer to Ref. [13].

## Acknowledgements

We like to thank Claude Bernard and Matthias Jamin for helpful discussions. DB, AK and KM would like to thank the IFAE at the Universitat Autònoma de Barcelona for hospitality. The work of DB is supported by the São Paulo Research Foundation (FAPESP) Grant No. 2015/20689-9 and by CNPq Grant No. 305431/2015-3. The work of MG is supported by the U.S. Department of Energy, Office of Science, Office of High Energy Physics, under Award Number DE-FG03-92ER40711. The work of AK is supported by STFC under the consolidated grant ST/N504130/1. KM is supported by a grant from the Natural Sciences and Engineering Research Council of Canada. The work of DN is supported by JSPS KAKENHI grant numbers JP16K05323 and JP17H01133. SP is supported by CICYTFEDER-FPA2014-55613-P, 2014-SGR-1450. The work of TT is supported by STFC under the consolidated grant ST/P000290/1.

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
