# Peer review of "The strong coupling from $e^+e^-\to$ hadrons"

_SciPost Physics Proceedings_

## Round 1 · Referee Report · Anonymous (Referee 1) · 2018-12-9

Report

This is an interesting and timely analysis. This contribution essentially summarises Ref. [4], to which the authors refer for details. However, I would like the authors to implement a few changes and improvements in the presentation and answer a few questions, see "Requested changes".

Requested changes

  1. Add captions to Tables and enumerate them.

  2. In section 2 the authors infer that more data in the region $s<4$ GeV${}^2$ should imply a more precise determination of $\alpha_s$ as compared with the higher $s$ region. However, I do not see why this should be true in general, since theoretical uncertainties play a crucial role in the extraction of $\alpha_s$.

  3. Explain briefly how the errors quoted in the last column of would-be Table 1 are obtained.

  4. The error on $\alpha_s$ displayed in Figure 4 is not the total error. This is stated by the authors in footnote 5, but it should also be stated in the caption of Figure 4. The total error should possibly be indicated in the Figure, or reported in the text.

  5. Data for $w_4$ should also be included in Figure 4, according to the content of would-be Table 2.

  6. What is the effect of neglected higher dimensional condensates on the data in Figure 4, and how do they compare with the duality violations (black circles) especially in the low $s_0^{min}$ region?

  7. The authors should address more clearly the stability of their fitted results reported in Figure 4 when varying $s_0^{min}$ and $s_0^{max}$. An analogous observation applies to section 5, see point 9.

  8. What is the channel displayed in Figure 5 left panel, V, A, V+A? And what are the uncertainties in the fitted curves?

  9. Section 5 concludes that the determination of $\alpha_s$ from $e^+e^-$ data and the one from $\tau$ data are consistent. However, there is hardly enough information that can be extracted from Figure 5 and the surrounding text. Importantly, what happens to the final value of $\alpha_s$ and to the fitted parameters of the duality-violation model when one varies $s_0^{min}$ in the $\tau$-data fits? This is a relevant point. I think that an accurate study of this dependence is needed in order to assess the stability and consistency of the results and to draw conclusions. If a complete analysis cannot be worked out in a reasonably short time, the authors can at least acknowledge this point in their contribution and formulate their final remarks accordingly.

---

## Editorial Decision

resubmitted